# DivideMix: Learning with Noisy Labels as Semi-supervised Learning

**Junnan Li, Richard Socher, Steven C.H. Hoi**
Salesforce Research
{junnan.li,rsocher,shoi}@salesforce.com

## Abstract

Deep neural networks are known to be annotation-hungry. Numerous efforts have been devoted to reducing the annotation cost when learning with deep networks. Two prominent directions include learning with noisy labels and semi-supervised learning by exploiting unlabeled data. In this work, we propose DivideMix, a novel framework for learning with noisy labels by leveraging semi-supervised learning techniques. In particular, DivideMix models the per-sample loss distribution with a mixture model to dynamically divide the training data into a labeled set with clean samples and an unlabeled set with noisy samples, and trains the model on both the labeled and unlabeled data in a semi-supervised manner. To avoid confirmation bias, we simultaneously train two diverged networks where each network uses the dataset division from the other network. During the semi-supervised training phase, we improve the MixMatch strategy by performing label co-refinement and label co-guessing on labeled and unlabeled samples, respectively. Experiments on multiple benchmark datasets demonstrate substantial improvements over state-of-the-art methods. Code is available at https://github.com/LiJunnan1992/DivideMix.

## 1 Introduction

The remarkable success in training deep neural networks (DNNs) is largely attributed to the collection of large datasets with human annotated labels. However, it is extremely expensive and time-consuming to label extensive data with high-quality annotations. On the other hand, there exist alternative and inexpensive methods for mining large-scale data with labels, such as querying commercial search engines (Li et al., 2017a), downloading social media images with tags (Mahajan et al., 2018), leveraging machine-generated labels (Kuznetsova et al., 2018), or using a single annotator to label each sample (Tanno et al., 2019). These alternative methods inevitably yield samples with *noisy labels*. A recent study (Zhang et al., 2017) shows that DNNs can easily overfit to noisy labels and results in poor generalization performance.

Existing methods on learning with noisy labels (LNL) primarily take a loss correction approach. Some methods estimate the noise transition matrix and use it to correct the loss function (Patrini et al., 2017; Goldberger & Ben-Reuven, 2017). However, correctly estimating the noise transition matrix is challenging. Some methods leverage the predictions from DNNs to correct labels and modify the loss accordingly (Reed et al., 2015; Tanaka et al., 2018). These methods do not perform well under high noise ratio as the predictions from DNNs would dominate training and cause overfitting. To overcome this, Arazo et al. (2019) adopt MixUp (Zhang et al., 2018) augmentation. Another approach selects or reweights samples so that noisy samples contribute less to the loss (Jiang et al., 2018; Ren et al., 2018). A challenging issue is to design a reliable criteria to select clean samples. It has been shown that DNNs tend to learn simple patterns first before fitting label noise (Arpit et al., 2017). Therefore, many methods treat samples with small loss as clean ones (Jiang et al., 2018; Arazo et al., 2019). Among those methods, Co-teaching (Han et al., 2018) and Co-teaching+ (Yu et al., 2019) train two networks where each network selects small-loss samples in a mini-batch to train the other.

Another active area of research that also aims to reduce annotation cost is semi-supervised learning (SSL). In SSL, the training data consists of unlabeled samples in addition to the labeled samples. Significant progress has been made in leveraging unlabeled samples by enforcing the model to produce

low entropy predictions on unlabeled data (Grandvalet & Bengio, 2004) or consistent predictions on perturbed input (Laine & Aila, 2017; Tarvainen & Valpola, 2017; Miyato et al., 2019). Recently, Berthelot et al. (2019) propose MixMatch, which unifies several dominant SSL approaches in one framework and achieves state-of-the-art performance.

Despite the individual advances in LNL and SSL, their connection has been underexplored. In this work, we propose DivideMix, which addresses learning with label noise in a semi-supervised manner. Different from most existing LNL approaches, DivideMix discards the sample labels that are highly likely to be noisy, and leverages the noisy samples as unlabeled data to regularize the model from overfitting and improve generalization performance. The key contributions of this work are:

- We propose co-divide, which trains two networks simultaneously. For each network, we dynamically fit a Gaussian Mixture Model (GMM) on its per-sample loss distribution to divide the training samples into a labeled set and an unlabeled set. The divided data is then used to train the *other* network. Co-divide keeps the two networks diverged, so that they can filter different types of error and avoid confirmation bias in self-training.

- During SSL phase, we improve MixMatch with label co-refinement and co-guessing to account for label noise. For labeled samples, we refine their ground-truth labels using the network's predictions guided by the GMM for the other network. For unlabeled samples, we use the ensemble of both networks to make reliable guesses for their labels.

- We experimentally show that DivideMix significantly advances state-of-the-art results on multiple benchmarks with different types and levels of label noise. We also provide extensive ablation study and qualitative results to examine the effect of different components.

## 2 RELATED WORK

### 2.1 LEARNING WITH NOISY LABELS

Most existing methods for training DNNs with noisy labels seek to correct the loss function. The correction can be categorized in two types. The first type treats all samples equally and correct loss either explicitly or implicitly through relabeling the noisy samples. For relabeling methods, the noisy samples are modeled with directed graphical models (Xiao et al., 2015), Conditional Random Fields (Vahdat, 2017), knowledge graph (Li et al., 2017b), or DNNs (Veit et al., 2017; Lee et al., 2018). However, they require access to a small set of clean samples. Recently, Tanaka et al. (2018) and Yi & Wu (2019) propose iterative methods which relabel samples using network predictions. For explicit loss correction. Reed et al. (2015) propose a bootstrapping method which modifies the loss with model predictions, and Ma et al. (2018) improve the bootstrapping method by exploiting the dimensionality of feature subspaces. Patrini et al. (2017) estimate the label corruption matrix for loss correction, and Hendrycks et al. (2018) improve the corruption matrix by using a clean set of data. The second type of correction focuses on reweighting training samples or separating clean and noisy samples, which results in correcting the loss function (Thulasidasan et al., 2019; Konstantinov & Lampert, 2019). A common method is to consider samples with smaller loss as clean ones (Shen & Sanghavi, 2019). Jiang et al. (2018) train a mentor network to guide a student network by assigning weights to samples. Ren et al. (2018) reweight samples based on their gradient directions. Chen et al. (2019) apply cross validation to identify clean samples. Arazo et al. (2019) calculate sample weights by modeling per-sample loss with a mixture model. Han et al. (2018) train two networks which select small-loss samples within each mini-batch to train each other, and Yu et al. (2019) improve it by updating the network on disagreement data to keep the two networks diverged.

Contrary to all aforementioned methods, our method discards the labels that are highly likely to be noisy, and utilize the noisy samples as unlabeled data to regularize training in a SSL manner. Ding et al. (2018) and Kong et al. (2019) have shown that SSL method is effective in LNL. However, their methods do not perform well under high levels of noise, whereas our method can better distinguish and utilize noisy samples. Besides leveraging SSL, our method also introduces other advantages. Compared to self-training methods (Jiang et al., 2018; Arazo et al., 2019), our method can avoid the confirmation bias problem (Tarvainen & Valpola, 2017) by training two networks to filter error for each other. Compared to Co-teaching (Han et al., 2018) and Co-teaching+ (Yu et al., 2019), our method is more robust to noise by enabling the two networks to teach each other implicitly at each epoch (co-divide) and explicitly at each mini-batch (label co-refinement and co-guessing).

Figure 1: DivideMix trains two networks (A and B) simultaneously. At each epoch, a network models its per-sample loss distribution with a GMM to divide the dataset into a labeled set (mostly clean) and an unlabeled set (mostly noisy), which is then used as training data for the other network (*i.e.* co-divide). At each mini-batch, a network performs semi-supervised training using an improved MixMatch method. We perform label co-refinement on the labeled samples and label co-guessing on the unlabeled samples.

## 2.2 SEMI-SUPERVISED LEARNING

SSL methods aim to improve the model's performance by leveraging unlabeled data. Current state-of-the-art SSL methods mostly involve adding an additional loss term on unlabeled data to regularize training. The regularization falls into two classes: consistency regularization (Laine & Aila, 2017; Tarvainen & Valpola, 2017; Miyato et al., 2019) enforces the model to produce consistent predictions on augmented input data; entropy minimization (Grandvalet & Bengio, 2004; Lee, 2013) encourages the model to give high-confidence predictions on unlabeled data. Recently, Berthelot et al. (2019) propose MixMatch, which unifies consistency regularization, entropy minimization, and the MixUp (Zhang et al., 2018) regularization into one framework.

## 3 METHOD

In this section, we introduce DivideMix, our proposed method for learning with noisy labels. An overview of the method is shown in Figure 1. To avoid confirmation bias of self-training where the model would accumulate its errors, we simultaneously train two networks to filter errors for each other through epoch-level implicit teaching and batch-level explicit teaching. At each epoch, we perform co-divide, where one network divides the noisy training dataset into a clean labeled set ($\mathcal{X}$) and a noisy unlabeled set ($\mathcal{U}$), which are then used by the other network. At each mini-batch, one network utilizes both labeled and unlabeled samples to perform semi-supervised learning guided by the other network. Algorithm 1 delineates the full algorithm.

### 3.1 CO-DIVIDE BY LOSS MODELING

Deep networks tend to learn clean samples faster than noisy samples (Arpit et al., 2017), leading to lower loss for clean samples (Han et al., 2018; Chen et al., 2019). Following Arazo et al. (2019), we aim to find the probability of a sample being clean by fitting a mixture model to the per-sample loss distribution. Formally, let $\mathcal{D} = (\mathcal{X}, \mathcal{Y}) = \{(x_i, y_i)\}_{i=1}^{N}$ denote the training data, where $x_i$ is an image and $y_i \in \{0, 1\}^C$ is the one-hot label over $C$ classes. Given a model with parameters $\theta$, the cross-entropy loss $\ell(\theta)$ reflects how well the model fits the training samples:

$$\ell(\theta) = \{\ell_i\}_{i=1}^{N} = \left\{ -\sum_{c=1}^{C} y_i^c \log(\mathrm{p}_{\mathrm{model}}^c(x_i; \theta)) \right\}_{i=1}^{N}, \tag{1}$$

where $\mathrm{p}_{\mathrm{model}}^c$ is the model's output softmax probability for class $c$.

Arazo et al. (2019) fit a two-component Beta Mixture Model (BMM) to the max-normalized loss $\ell$ to model the distribution of clean and noisy samples. However, we find that BMM tends to produce undesirable flat distributions and fails when the label noise is asymmetric. Instead, Gaussian Mixture Model (GMM) (Permuter et al., 2006) can better distinguish clean and noisy samples due to its flexibility in the sharpness of distribution. Therefore, we fit a two-component GMM to $\ell$ using the Expectation-Maximization algorithm. For each sample, its clean probability $w_i$ is the posterior probability $p(g|\ell_i)$, where $g$ is the Gaussian component with smaller mean (smaller loss).

We divide the training data into a labeled set and an unlabeled set by setting a threshold $\tau$ on $w_i$. However, training a model using the data divided by itself could lead to confirmation bias (*i.e.* the

---

**Algorithm 1:** DivideMix. Line 4-8: co-divide; Line 17-18: label co-refinement; Line 20: label co-guessing.

---

1 **Input:** $\theta^{(1)}$ and $\theta^{(2)}$, training dataset $(\mathcal{X}, \mathcal{Y})$, clean probability threshold $\tau$, number of augmentations $M$, sharpening temperature $T$, unsupervised loss weight $\lambda_u$, Beta distribution parameter $\alpha$ for MixMatch.

2 $\theta^{(1)}, \theta^{(2)} = \text{WarmUp}(\mathcal{X}, \mathcal{Y}, \theta^{(1)}, \theta^{(2)})$            *// standard training (with confidence penalty)*

3 **while** $e < \text{MaxEpoch}$ **do**

4      $\mathcal{W}^{(2)} = \text{GMM}(\mathcal{X}, \mathcal{Y}, \theta^{(1)})$        *// model per-sample loss with $\theta^{(1)}$ to obtain clean proability for $\theta^{(2)}$*

5      $\mathcal{W}^{(1)} = \text{GMM}(\mathcal{X}, \mathcal{Y}, \theta^{(2)})$        *// model per-sample loss with $\theta^{(2)}$ to obtain clean proability for $\theta^{(1)}$*

6      **for** $k = 1, 2$ **do**            *// train the two networks one by one*

7          $\mathcal{X}_e^{(k)} = \{(x_i, y_i, w_i) | w_i \geq \tau, \forall (x_i, y_i, w_i) \in (\mathcal{X}, \mathcal{Y}, \mathcal{W}^{(k)})\}$     *// labeled training set for $\theta^{(k)}$*

8          $\mathcal{U}_e^{(k)} = \{x_i | w_i < \tau, \forall (x_i, w_i) \in (\mathcal{X}, \mathcal{W}^{(k)})\}$       *// unlabeled training set for $\theta^{(k)}$*

9          **for** iter $= 1$ **to** num_iters **do**

10             From $\mathcal{X}_e^{(k)}$, draw a mini-batch $\{(x_b, y_b, w_b); b \in (1, ..., B)\}$

11             From $\mathcal{U}_e^{(k)}$, draw a mini-batch $\{u_b; b \in (1, ..., B)\}$

12             **for** $b = 1$ **to** $B$ **do**

13                **for** $m = 1$ **to** $M$ **do**

14                   $\hat{x}_{b,m} = \text{Augment}(x_b)$        *// apply $m^{th}$ round of augmentation to $x_b$*

15                   $\hat{u}_{b,m} = \text{Augment}(u_b)$        *// apply $m^{th}$ round of augmentation to $u_b$*

16                **end**

17                $p_b = \frac{1}{M} \sum_m p_{\text{model}}(\hat{x}_{b,m}; \theta^{(k)})$     *// average the predictions across augmentations of $x_b$*

18                $\bar{y}_b = w_b y_b + (1 - w_b) p_b$

                   *// refine ground-truth label guided by the clean probability produced by the other network*

19                $\hat{y}_b = \text{Sharpen}(\bar{y}_b, T)$        *// apply temperature sharpening to the refined label*

20                $\bar{q}_b = \frac{1}{2M} \sum_m \left(p_{\text{model}}(\hat{u}_{b,m}; \theta^{(1)}) + p_{\text{model}}(\hat{u}_{b,m}; \theta^{(2)})\right)$

                   *// co-guessing: average the predictions from both networks across augmentations of $u_b$*

21                $q_b = \text{Sharpen}(\bar{q}_b, T)$        *// apply temperature sharpening to the guessed label*

22             **end**

23             $\hat{\mathcal{X}} = \{(\hat{x}_{b,m}, \hat{y}_b); b \in (1, ..., B), m \in (1, ..., M)\}$     *// augmented labeled mini-batch*

24             $\hat{\mathcal{U}} = \{(\hat{u}_{b,m}, q_b); b \in (1, ..., B), m \in (1, ..., M)\}$     *// augmented unlabeled mini-batch*

25             $\mathcal{L}_\mathcal{X}, \mathcal{L}_\mathcal{U} = \text{MixMatch}(\hat{\mathcal{X}}, \hat{\mathcal{U}})$        *// apply MixMatch*

26             $\mathcal{L} = \mathcal{L}_\mathcal{X} + \lambda_u \mathcal{L}_\mathcal{U} + \lambda_r \mathcal{L}_{\text{reg}}$        *// total loss*

27             $\theta^{(k)} = \text{SGD}(\mathcal{L}, \theta^{(k)})$        *// update model parameters*

28          **end**

29      **end**

30 **end**

---

model is prone to confirm its mistakes (Tarvainen & Valpola, 2017)), as noisy samples that are wrongly grouped into the labeled set would keep having lower loss due to the model overfitting to their labels. Therefore, we propose co-divide to avoid error accumulation. In co-divide, the GMM for one network is used to divide training data for the other network. The two networks are kept diverged from each other due to different (random) parameter initialization, different training data division, different (random) mini-batch sequence, and different training targets. Being diverged offers the two networks distinct abilities to filter different types of error, making the model more robust to noise.

**Confidence Penalty for Asymmetric Noise.** For initial convergence of the algorithm, we need to "warm up" the model for a few epochs by training on all data using the standard cross-entropy loss. The warm up is effective for symmetric (*i.e.* uniformly random) label noise. However, for asymmetric (*i.e.* class-conditional) label noise, the network would quickly overfit to noise during warm up and produce over-confident (low entropy) predictions, which leads to most samples having near-zero normalized loss (see Figure 2a). In such cases, the GMM cannot effectively distinguish clean and noisy samples based on the loss distribution. To address this issue, we penalize confident predictions from the network by adding a negative entropy term, $-\mathcal{H}$ (Pereyra et al., 2017), to the cross-entropy loss during warm up. The entropy of a model's prediction for an input $x$ is defined as:

$$\mathcal{H} = -\sum_c p_{\text{model}}^c(x; \theta) \log(p_{\text{model}}^c(x; \theta)), \tag{2}$$

By maximizing the entropy, $\ell$ becomes more evenly distributed (see Figure 2b) and easier to be modeled by the GMM. Furthermore, in Figure 2c we show $\ell$ when the model is trained with

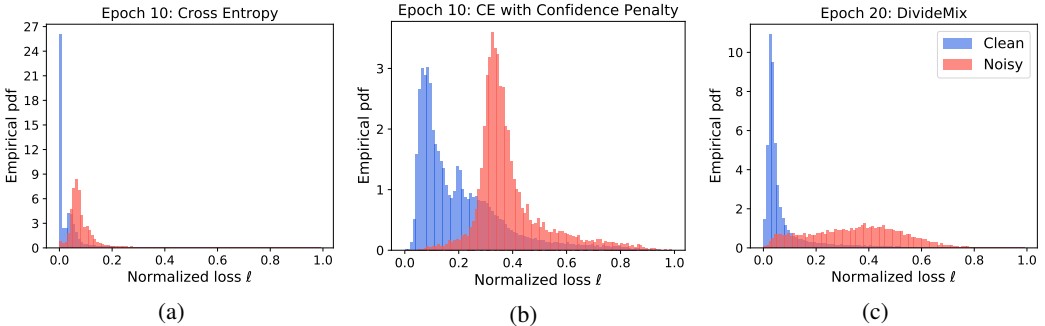

Figure 2: Training on CIFAR-10 with 40% asymmetric noise, warm up for 10 epochs. (a) Standard training with cross-entropy loss causes the model to overfit and produce over-confident predictions, making $\ell$ difficult to be modeled by the GMM. (b) Adding a confidence penalty (negative entropy) during warm up leads to more evenly-distributed $\ell$. (c) Training with DivideMix can effectively reduce the loss for clean samples while keeping the loss larger for most noisy samples.

DivideMix for 10 more epochs after warm up. The proposed method can significantly reduce the loss for clean samples while keeping the loss larger for most noisy samples.

### 3.2 MixMatch with Label Co-Refinement and Co-Guessing

At each epoch, having divided the training data, we train the two networks one at a time while keeping the other one fixed. Given a mini-batch of labeled samples with their corresponding one-hot labels and clean probability, $\{(x_b, y_b, w_b); b \in (1, ..., B)\}$, and a mini-batch of unlabeled samples $\{u_b; b \in (1, ..., B)\}$, we exploit MixMatch (Berthelot et al., 2019) for SSL. MixMatch utilizes unlabeled data by merging consistency regularization (*i.e.* encourage the model to output same predictions on perturbed unlabeled data) and entropy minimization (*i.e.* encourage the model to output confident predictions on unlabeled data) with the MixUp (Zhang et al., 2018) augmentation (*i.e.* encourage the model to have linear behaviour between samples).

To account for label noise, we make two improvements to MixMatch which enable the two networks to teach each other. First, we perform label co-refinement for labeled samples by linearly combining the ground-truth label $y_b$ with the network's prediction $p_b$ (averaged across multiple augmentations of $x_b$), guided by the clean probability $w_b$ produced by the other network:

$$\bar{y}_b = w_b y_b + (1 - w_b) p_b. \tag{3}$$

Then we apply a sharpening function on the refined label to reduce its temperature:

$$\hat{y}_b = \text{Sharpen}(\bar{y}_b, T) = \bar{y}_b^{c \frac{1}{T}} \Big/ \sum_{c=1}^{C} \bar{y}_b^{c \frac{1}{T}}, \text{ for } c = 1, 2, ..., C. \tag{4}$$

Second, we use the ensemble of predictions from both networks to "co-guess" the labels for unlabeled samples (algorithm 1, line 20), which can produce more reliable guessed labels.

Having acquired $\hat{\mathcal{X}}$ (and $\hat{\mathcal{U}}$) which consists of multiple augmentations of labeled (unlabeled) samples and their refined (guessed) labels, we follow MixMatch to "mix" the data, where each sample is interpolated with another sample randomly chosen from the combined mini-batch of $\hat{\mathcal{X}}$ and $\hat{\mathcal{U}}$. Specifically, for a pair of samples $(x_1, x_2)$ and their corresponding labels $(p_1, p_2)$, the mixed $(x', p')$ is computed by:

$$\lambda \sim \text{Beta}(\alpha, \alpha), \tag{5}$$
$$\lambda' = \max(\lambda, 1 - \lambda), \tag{6}$$
$$x' = \lambda' x_1 + (1 - \lambda') x_2, \tag{7}$$
$$p' = \lambda' p_1 + (1 - \lambda') p_2. \tag{8}$$

MixMatch transforms $\hat{\mathcal{X}}$ and $\hat{\mathcal{U}}$ into $\mathcal{X}'$ and $\mathcal{U}'$. Equation 6 ensures that $\mathcal{X}'$ are "closer" to $\hat{\mathcal{X}}$ than $\hat{\mathcal{U}}$. The loss on $\mathcal{X}'$ is the cross-entropy loss and the loss on $\mathcal{U}'$ is the mean squared error:

$$\mathcal{L}_{\mathcal{X}} = -\frac{1}{|\mathcal{X}'|} \sum_{x,p \in \mathcal{X}'} \sum_c p_c \log(\mathrm{p}_{\mathrm{model}}^c(x;\theta)), \tag{9}$$

$$\mathcal{L}_{\mathcal{U}} = \frac{1}{|\mathcal{U}'|} \sum_{x,p \in \mathcal{U}'} \|p - \mathrm{p}_{\mathrm{model}}(x;\theta)\|_2^2. \tag{10}$$

Under high levels of noise, the network would be encouraged to predict the same class to minimize the loss. To prevent assigning all samples to a single class, we apply the regularization term used by Tanaka et al. (2018) and Arazo et al. (2019), which uses a uniform prior distribution $\pi$ (i.e. $\pi_c = 1/C$) to regularize the model's average output across all samples in the mini-batch:

$$\mathcal{L}_{\mathrm{reg}} = \sum_c \pi_c \log \left( \pi_c \bigg/ \frac{1}{|\mathcal{X}'| + |\mathcal{U}'|} \sum_{x \in \mathcal{X}' + \mathcal{U}'} \mathrm{p}_{\mathrm{model}}^c(x;\theta) \right). \tag{11}$$

Finally, the total loss is:

$$\mathcal{L} = \mathcal{L}_{\mathcal{X}} + \lambda_u \mathcal{L}_{\mathcal{U}} + \lambda_r \mathcal{L}_{\mathrm{reg}}. \tag{12}$$

In our experiments, we set $\lambda_r$ as 1 and use $\lambda_u$ to control the strength of the unsupervised loss.

## 4 EXPERIMENTS

### 4.1 DATASETS AND IMPLEMENTATION DETAILS

We extensively validate our method on four benchmark datasets, namely CIFAR-10, CIFAR-100 (Krizhevsky & Hinton, 2009), Clothing1M (Xiao et al., 2015), and WebVision (Li et al., 2017a). Both CIFAR-10 and CIFAR-100 contain 50K training images and 10K test images of size $32 \times 32$. Following previous works (Tanaka et al., 2018; Li et al., 2019), we experiment with two types of label noise: *symmetric* and *asymmetric*. Symmetric noise is generated by randomly replacing the labels for a percentage of the training data with all possible labels. Note that there is another criterion for symmetric label noise injection where the true labels cannot be maintained (Jiang et al., 2018; Wang et al., 2018), for which we also report the results (Table 6 in Appendix). Asymmetric noise is designed to mimic the structure of real-world label noise, where labels are only replaced by similar classes (e.g. deer→horse, dog↔cat).

We use an 18-layer PreAct Resnet (He et al., 2016) and train it using SGD with a momentum of 0.9, a weight decay of 0.0005, and a batch size of 128. The network is trained for 300 epochs. We set the initial learning rate as 0.02, and reduce it by a factor of 10 after 150 epochs. The warm up period is 10 epochs for CIFAR-10 and 30 epochs for CIFAR-100. We find that most hyperparameters introduced by DivideMix do not need to be heavily tuned. For all CIFAR experiments, we use the same hyperparameters $M = 2$, $T = 0.5$, and $\alpha = 4$. $\tau$ is set as 0.5 except for $90\%$ noise ratio when it is set as 0.6. We choose $\lambda_u$ from $\{0, 25, 50, 150\}$ using a small validation set.

Clothing1M and WebVision 1.0 are two large-scale datasets with real-world noisy labels. Clothing1M consists of 1 million training images collected from online shopping websites with labels generated from surrounding texts. We follow previous work (Li et al., 2019) and use ResNet-50 with ImageNet pretrained weights. WebVision contains 2.4 million images crawled from the web using the 1,000 concepts in ImageNet ILSVRC12. Following previous work (Chen et al., 2019), we compare baseline methods on the first 50 classes of the Google image subset using the inception-resnet v2 (Szegedy et al., 2017). The training details are delineated in Appendix B.

### 4.2 COMPARISON WITH STATE-OF-THE-ART METHODS

We compare DivideMix with multiple baselines using the same network architecture. Here we introduce some of the most recent state-of-the-art methods: Meta-Learning (Li et al., 2019) proposes a gradient based method to find model parameters that are more noise-tolerant; Joint-Optim (Tanaka et al., 2018) and P-correction (Yi & Wu, 2019) jointly optimize the sample labels and the network parameters; M-correction (Arazo et al., 2019) models sample loss with BMM and applies MixUp.

| Dataset | | CIFAR-10 | | | | CIFAR-100 | | | |
|---|---|---|---|---|---|---|---|---|---|
| Method/Noise ratio | | 20% | 50% | 80% | 90% | 20% | 50% | 80% | 90% |
| Cross-Entropy | Best | 86.8 | 79.4 | 62.9 | 42.7 | 62.0 | 46.7 | 19.9 | 10.1 |
| | Last | 82.7 | 57.9 | 26.1 | 16.8 | 61.8 | 37.3 | 8.8 | 3.5 |
| Bootstrap (Reed et al., 2015) | Best | 86.8 | 79.8 | 63.3 | 42.9 | 62.1 | 46.6 | 19.9 | 10.2 |
| | Last | 82.9 | 58.4 | 26.8 | 17.0 | 62.0 | 37.9 | 8.9 | 3.8 |
| F-correction (Patrini et al., 2017) | Best | 86.8 | 79.8 | 63.3 | 42.9 | 61.5 | 46.6 | 19.9 | 10.2 |
| | Last | 83.1 | 59.4 | 26.2 | 18.8 | 61.4 | 37.3 | 9.0 | 3.4 |
| Co-teaching+* (Yu et al., 2019) | Best | 89.5 | 85.7 | 67.4 | 47.9 | 65.6 | 51.8 | 27.9 | 13.7 |
| | Last | 88.2 | 84.1 | 45.5 | 30.1 | 64.1 | 45.3 | 15.5 | 8.8 |
| Mixup (Zhang et al., 2018) | Best | 95.6 | 87.1 | 71.6 | 52.2 | 67.8 | 57.3 | 30.8 | 14.6 |
| | Last | 92.3 | 77.6 | 46.7 | 43.9 | 66.0 | 46.6 | 17.6 | 8.1 |
| P-correction* (Yi & Wu, 2019) | Best | 92.4 | 89.1 | 77.5 | 58.9 | 69.4 | 57.5 | 31.1 | 15.3 |
| | Last | 92.0 | 88.7 | 76.5 | 58.2 | 68.1 | 56.4 | 20.7 | 8.8 |
| Meta-Learning* (Li et al., 2019) | Best | 92.9 | 89.3 | 77.4 | 58.7 | 68.5 | 59.2 | 42.4 | 19.5 |
| | Last | 92.0 | 88.8 | 76.1 | 58.3 | 67.7 | 58.0 | 40.1 | 14.3 |
| M-correction (Arazo et al., 2019) | Best | 94.0 | 92.0 | 86.8 | 69.1 | 73.9 | 66.1 | 48.2 | 24.3 |
| | Last | 93.8 | 91.9 | 86.6 | 68.7 | 73.4 | 65.4 | 47.6 | 20.5 |
| DivideMix | Best | **96.1** | **94.6** | **93.2** | **76.0** | **77.3** | **74.6** | **60.2** | **31.5** |
| | Last | **95.7** | **94.4** | **92.9** | **75.4** | **76.9** | **74.2** | **59.6** | **31.0** |

Table 1: Comparison with state-of-the-art methods in test accuracy (%) on CIFAR-10 and CIFAR-100 with symmetric noise. Methods marked by * denote re-implementations based on public code.

Note that none of these methods can consistently outperform others across different datasets. M-correction excels at symmetric noise, whereas Meta-Learning performs better for asymmetric noise.

Table 1 shows the results on CIFAR-10 and CIFAR-100 with different levels of symmetric label noise ranging from $20\%$ to $90\%$. We report both the best test accuracy across all epochs and the averaged test accuracy over the last 10 epochs. DivideMix outperforms state-of-the-art methods by a large margin across all noise ratios. The improvement is substantial ($\sim 10\%$ in accuracy) for the more challenging CIFAR-100 with high noise ratios. Appendix A shows comparison with more methods in Table 6. The results on CIFAR-10 with asymmetric noise is shown in Table 2. We use $40\%$ because certain classes become theoretically indistinguishable for asymmetric noise larger than $50\%$.

| Method | | Best | Last |
|---|---|---|---|
| Cross-Entropy | | 85.0 | 72.3 |
| F-correction (Patrini et al., 2017) | | 87.2 | 83.1 |
| M-correction (Arazo et al., 2019) | | 87.4 | 86.3 |
| Iterative-CV (Chen et al., 2019) | | 88.6 | 88.0 |
| P-correction (Yi & Wu, 2019) | | 88.5 | 88.1 |
| Joint-Optim (Tanaka et al., 2018) | | 88.9 | 88.4 |
| Meta-Learning (Li et al., 2019) | | 89.2 | 88.6 |
| DivideMix | | **93.4** | **92.1** |

Table 2: Comparison with state-of-the-art methods in test accuracy (%) on CIFAR-10 with 40% asymmetric noise. We re-implement all methods under the same setting.

Table 3 and Table 4 show the results on Clothing1M and WebVision, respectively. DivideMix consistently outperforms state-of-the-art methods across all datasets with different types of label noise. For WebVision, we achieve more than 12% improvement in top-1 accuracy.

| Method | Test Accuracy |
|---|---|
| Cross-Entropy | 69.21 |
| F-correction (Patrini et al., 2017) | 69.84 |
| M-correction (Arazo et al., 2019) | 71.00 |
| Joint-Optim (Tanaka et al., 2018) | 72.16 |
| Meta-Cleaner (Zhang et al., 2019) | 72.50 |
| Meta-Learning (Li et al., 2019) | 73.47 |
| P-correction (Yi & Wu, 2019) | 73.49 |
| DivideMix | **74.76** |

Table 3: Comparison with state-of-the-art methods in test accuracy (%) on Clothing1M. Results for baselines are copied from original papers.

| Method | WebVision | | ILSVRC12 | |
|---|---|---|---|---|
| | top1 | top5 | top1 | top5 |
| F-correction (Patrini et al., 2017) | 61.12 | 82.68 | 57.36 | 82.36 |
| Decoupling (Malach & Shalev-Shwartz, 2017) | 62.54 | 84.74 | 58.26 | 82.26 |
| D2L (Ma et al., 2018) | 62.68 | 84.00 | 57.80 | 81.36 |
| MentorNet (Jiang et al., 2018) | 63.00 | 81.40 | 57.80 | 79.92 |
| Co-teaching (Han et al., 2018) | 63.58 | 85.20 | 61.48 | 84.70 |
| Iterative-CV (Chen et al., 2019) | 65.24 | 85.34 | 61.60 | 84.98 |
| DivideMix | **77.32** | **91.64** | **75.20** | **90.84** |

Table 4: Comparison with state-of-the-art methods trained on (mini) WebVision dataset. Numbers denote top-1 (top-5) accuracy (%) on the WebVision validation set and the ImageNet ILSVRC12 validation set. Results for baseline methods are copied from Chen et al. (2019).

## 4.3 ABLATION STUDY

We study the effect of removing different components to provide insights into what makes DivideMix successful. We analyze the results in Table 5 as follows. Appendix C contains additional explanations.

| Dataset | | CIFAR-10 | | | | | CIFAR-100 | | | |
|---|---|---|---|---|---|---|---|---|---|---|
| Noise type | | Sym. | | | | Asym. | Sym. | | | |
| Methods/Noise ratio | | 20% | 50% | 80% | 90% | 40% | 20% | 50% | 80% | 90% |
| DivideMix | Best | **96.1** | **94.6** | **93.2** | **76.0** | **93.4** | **77.3** | **74.6** | **60.2** | **31.5** |
| | Last | **95.7** | **94.4** | **92.9** | **75.4** | **92.1** | **76.9** | **74.2** | **59.6** | **31.0** |
| DivideMix with $\theta^{(1)}$ test | Best | 95.2 | 94.2 | 93.0 | 75.5 | 92.7 | 75.2 | 72.8 | 58.3 | 29.9 |
| | Last | 95.0 | 93.7 | 92.4 | 74.2 | 91.4 | 74.8 | 72.1 | 57.6 | 29.2 |
| DivideMix w/o co-training | Best | 95.0 | 94.0 | 92.6 | 74.3 | 91.9 | 74.8 | 72.3 | 56.7 | 27.7 |
| | Last | 94.8 | 93.3 | 92.2 | 73.2 | 90.6 | 74.1 | 71.7 | 56.3 | 27.2 |
| DivideMix w/o label refinement | Best | 96.0 | 94.6 | 93.0 | 73.7 | 87.7 | 76.9 | 74.2 | 58.7 | 26.9 |
| | Last | 95.5 | 94.2 | 92.7 | 73.0 | 86.3 | 76.4 | 73.9 | 58.2 | 26.3 |
| DivideMix w/o augmentation | Best | 95.3 | 94.1 | 92.2 | 73.9 | 89.5 | 76.5 | 73.1 | 58.2 | 26.9 |
| | Last | 94.9 | 93.5 | 91.8 | 73.0 | 88.4 | 76.2 | 72.6 | 58.0 | 26.4 |
| Divide and MixMatch | Best | 94.1 | 92.8 | 89.7 | 70.1 | 86.5 | 73.7 | 70.5 | 55.3 | 25.0 |
| | Last | 93.5 | 92.3 | 89.1 | 68.6 | 85.2 | 72.4 | 69.7 | 53.9 | 23.7 |

Table 5: Ablation study results in terms of test accuracy (%) on CIFAR-10 and CIFAR-100.

- To study the effect of model ensemble during *test*, we use the prediction from a single model $\theta^{(1)}$ instead of averaging the predictions from both networks as in DivideMix. Note that the training process remains unchanged. The decrease in accuracy suggests that the ensemble of two diverged networks consistently yields better performance during inference.
- To study the effect of co-training, we train a single network using self-divide (*i.e.* divide the training data based on its own loss). The performance further decreases compared to $\theta^{(1)}$.

- We find that both label refinement and input augmentation are beneficial for DivideMix.
- We combine self-divide with the original MixMatch as a naive baseline for using SLL in LNL.

Appendix A also introduces more in-depth studies in examining the robustness of our method to label noise, including the AUC for clean/noisy sample classification on CIFAR-10 training data, qualitative examples from Clothing1M where our method can effectively identify the noisy samples and leverage them as unlabeled data, and visualization results using t-SNE.

## 5 CONCLUSION

In this paper, we propose DivideMix for learning with noisy labels by leveraging SSL. Our method trains two networks simultaneously and achieves robustness to noise through dataset co-divide, label co-refinement and co-guessing. Through extensive experiments across multiple datasets, we show that DivideMix consistently exhibits substantial performance improvements compared to state-of-the-art methods. For future work, we are interested in incorporating additional ideas from SSL to LNL, and vice versa. Furthermore, we are also interested in adapting DivideMix to other domains such as NLP.

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

## APPENDIX A  ADDITIONAL EXPERIMENT RESULTS

In Table 6, we compare DivideMix with previous methods under the symmetric noise setting where true labels cannot be maintained. DivideMix significantly outperforms previous methods which use deeper or wider network architectures.

| Method | Architecture | CIFAR-10 | | | | CIFAR-100 | | | |
|---|---|---|---|---|---|---|---|---|---|
| | | 20% | 40% | 60% | 80% | 20% | 40% | 60% | 80% |
| MentorNet (Jiang et al., 2018) | WRN-101 | 92.0 | 89.0 | - | 49.0 | 73.0 | 68.0 | - | 35.0 |
| D2L (Ma et al., 2018) | CNN-12/RN-44 | 85.1 | 83.4 | 72.8 | - | 62.2 | 52.0 | 42.3 | - |
| Reweight (Ren et al., 2018) | WRN-28 | 86.9 | - | - | - | 61.3 | - | - | - |
| Abstention (Thulasidasan et al., 2019) | WRN-28 | 93.4 | 90.9 | 87.6 | 70.8 | 75.8 | 68.2 | 59.4 | 34.1 |
| DivideMix | PRN-18 | **96.2** | **94.9** | **94.3** | **79.8** | **77.2** | **75.2** | **72.0** | **60.0** |

Table 6: Comparison with state-of-the-art methods in test accuracy (%) on CIFAR-10 and CIFAR-100 with symmetric noise. Numbers are copied from original papers. Key: WRN (Wide ResNet), PRN (PreActivation ResNet). DivideMix outperforms previous methods that use deeper/wider networks.

In Figure 3, we show the Area Under a Curve (AUC) for clean/noisy sample classification on CIFAR-10 training data from one of the GMMs during the first 100 epochs. Our method can effectively separate clean and noisy samples as training proceeds, even for high noise ratio.

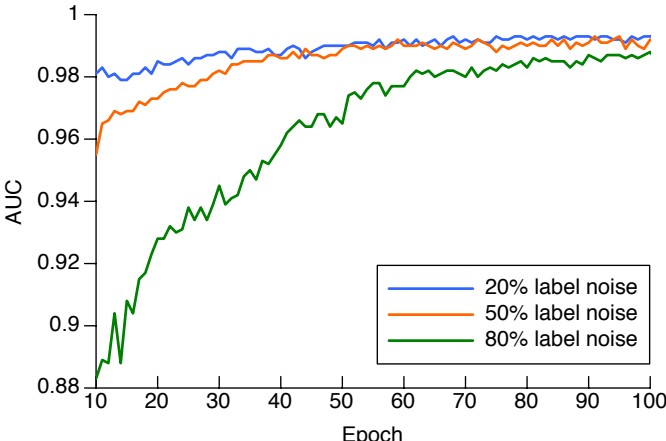

Figure 3: Area Under a Curve for clean/noisy image classification on CIFAR-10 training samples. Our method can effectively filter out the noisy samples and leverage them as unlabeled data.

In Figure 4, we show example images in Clothing1M identified by our method as noisy samples. Our method achieves noise filtering by discarding the noisy labels (shown in red) and using the co-guessed labels (shown in blue) to regularize training.

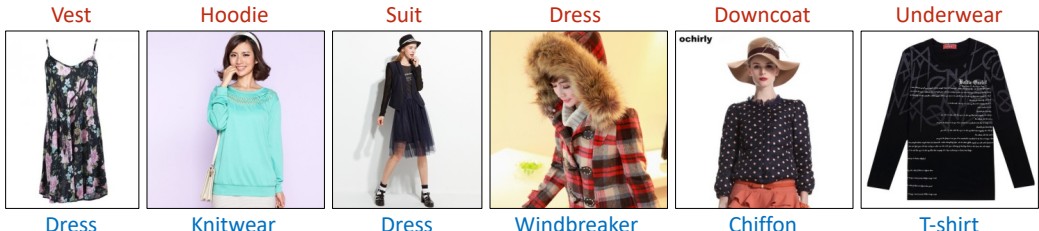

Figure 4: Clothing1M images identified as noisy samples by our method. Ground-truth labels are shown above in red and the co-guessed labels are shown below in blue.

In Figure 5, we visualize the features of training images using t-SNE (Maaten & Hinton, 2008). The model is trained using DivideMix for 200 epochs on CIFAR-10 with 80% label noise. The embeddings form 10 distinct clusters corresponding to the true class labels, not the noisy training labels, which demonstrates our method's robustness to label noise.

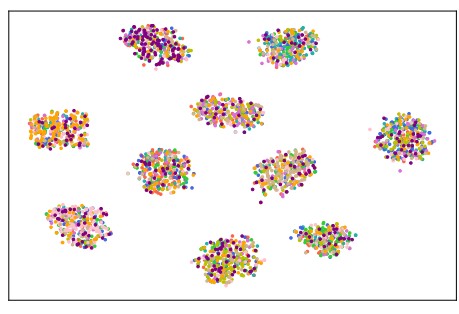 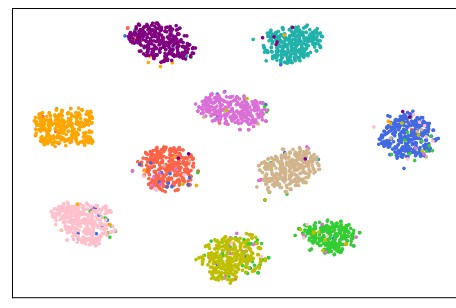

(a) Noisy training labels.      (b) True labels.

Figure 5: T-SNE of training images after training the model using DivideMix for 200 epochs on CIFAR-10 with 80% label noise. Different colors indicate (a) noisy training labels or (b) true labels. DivideMix is able to learn the true class distribution of the training data despite the label noise.

## APPENDIX B    ADDITIONAL TRAINING DETAILS

For CIFAR experiments, the only hyperparameter that we tune on a per-experiment basis is the unsupervised loss weight $\lambda_u$. Table 7 shows the value that we use. A larger $\lambda_u$ is required for stronger regularization under high noise ratios or with more classes.

For both Clothing1M and WebVision, we use the same set of hyperparameters $M = 2$, $T = 0.5$, $\tau = 0.5$, $\lambda_u = 0$, $\alpha = 0.5$, and train the network using SGD with a momentum of 0.9, a weight decay of 0.001, and a batch size of 32. The warm up period is 1 epoch. For Clothing1M, we train the network for 80 epochs. The initial learning rate is set as 0.002 and reduced by a factor of 10 after 40 epochs. For each epoch, we sample 1000 mini-batches from the training data while ensuring the labels (noisy) are balanced. For WebVision, we train the network for 100 epochs. The initial learning rate is set as 0.01 and reduced by a factor of 10 after 50 epochs.

| Hyperparameter | CIFAR-10 | | | | | CIFAR-100 | | | |
|---|---|---|---|---|---|---|---|---|---|
| | Asym. 40% | 20% | 50% | 80% | 90% | 20% | 50% | 80% | 90% |
| $\lambda_u$ | 0 | 0 | 25 | 25 | 50 | 25 | 150 | 150 | 150 |

Table 7: Unsupervised loss weight $\lambda_u$ for CIFAR experiments. Higher noise ratio requires stronger regularization from unlabeled samples.

## APPENDIX C    ADDITIONAL EXPLANATIONS FOR ABLATION STUDY

Here we clarify some details for the baseline methods in the ablation study. First, *DivideMix w/o co-training* still has dataset division, label refinement and label guessing, but performed by the same model. Thus, the performance drop (especially for CIFAR-100 with high noise ratio) suggests the disadvantage of self-training. Second, *label refinement* is important for high noise ratio because more noisy samples would be mistakenly divided into the labeled set. Third, *augmentation* improves performance through both producing more reliable predictions and achieving consistency regularization. In addition, same as Berthelot et al. (2019), we also find that *temperature sharpening* is essential for our method to perform well.

## APPENDIX D  TRAINING TIME ANALYSIS

We analyse the training time of DivideMix to understand its efficiency. In Table 8, we compare the total training time of DivideMix on CIFAR-10 with several state-of-the-art methods, using a single Nvidia V100 GPU. DivideMix is slower than Co-teaching+ (Yu et al., 2019), but faster than P-correction (Yi & Wu, 2019) and Meta-Learning (Li et al., 2019) which involve multiple training iterations. In Table 9, we also break down the computation time for each operation in DivideMix.

| Co-teaching+[*] | P-correction | Meta-Learning | DivideMix |
|:---:|:---:|:---:|:---:|
| 4.3 h | 6.0 h | 8.6 h | 5.2 h |

Table 8: Comparison of total training time (hours) on CIFAR-10.

| Co-Divide (Alg. 1, line 4-8) | Data MixMatch (Alg. 1, line 12-24) | Forward-Backward (Alg. 1, line 25-27) |
|:---:|:---:|:---:|
| 17.2 s | 16.0 s | 12.5 s |

Table 9: Computation time (seconds) per-epoch for each operation in DivideMix during training.

