# OpenReview forum: "DivideMix: Learning with Noisy Labels as Semi-supervised Learning"
_ICLR.cc/2020/Conference — Accept (Poster)_

### Official Review · AnonReviewer2 · 2019-10-16
**Official Blind Review #2**

**Rating:** 6

**Review:**

This paper proposed a method named DivideMix for learning with noisy labels, on top of the recent semi-supervised learning method MixMatch from Google. The idea is to model the per-sample loss distribution with a mixture model to dynamically DIVIDE the training data into (a labeled set with clean samples) and (an unlabeled set with noisy samples) and trains the model on both the labeled and unlabeled data in a semi-supervised manner.

The novelty is borderline. In the area of learning with noisy labels, it is known that SSL can work under this problem setting for several years, for example, the famous method "virtual adversarial training" from ICLR 2016 and a recent method "smooth neighbors on teacher graphs" from CVPR 2018. As a consequence, it is not surprising that the latest MixMatch can work as well, since MixMatch comes from mixup, virtual adversarial training and entropy minimization. This makes the novelty borderline. However, the significance may still be high according to the reported experimental results, and thus we may accept it to let more deep learning practitioners see the promising results.

**Experience Assessment:**

I have published in this field for several years.

**Review Assessment: Checking Correctness Of Derivations And Theory:**

I assessed the sensibility of the derivations and theory.

**Review Assessment: Checking Correctness Of Experiments:**

I assessed the sensibility of the experiments.

**Review Assessment: Thoroughness In Paper Reading:**

I read the paper thoroughly.

---

> ### Author Response · Authors · 2019-11-15
> **Response to Reviewer #2**
>
> We thank Reviewer 2 for the insightful comments. We believe that our work can inspire new research ideas to explore the intersection between the areas of learning with noisy labels and semi-supervised learning.

---

### Official Review · AnonReviewer1 · 2019-10-23
**Official Blind Review #1**

**Rating:** 6

**Review:**

This paper proposes an algorithm that learns with noisy labels that achieves state-of-the-art results. Their algorithm tries to exploit the noisy samples by assigning a ‘correct’ label through MixMatch. It borrows the idea from both semi-supervised learning and learning with label noise.

Suggestions:

1. When the author talked about “correct the loss function”, there are in fact two types of corrections: the first type tries to correct the loss function by equally treating all the samples, e.g., classical Huber loss, or F-correction. The second type tries to either re-weight samples or separate clean and noisy samples explicitly, which also results in correcting the loss function. It would be more clear if Section 2.1 can emphasize the difference between the two.

2. Also, there are other papers providing different but insightful ideas to label noise problem. The authors addressed the idea of using co-learning to avoid confirmation bias. However, it should be noted that this is not the only way to avoid confirmation bias, and their are other methods without using two networks [1-2], both providing some theoretical insights to the problem. It would be good to include them in the related work as well.

3. I would like to see a comparison of running time other than the accuracy, understanding the efficiency of each algorithm is important from a practical perspective.

[1] Learning with Bad Training Data via Iterative Trimmed Loss Minimization, Yanyao Shen, Sujay Sanghavi, ICML 2019.
[2] Robust Learning from Untrusted Sources,Nikola Konstantinov, Christoph H. Lampert, ICML 2019



**Experience Assessment:**

I have published one or two papers in this area.

**Review Assessment: Checking Correctness Of Derivations And Theory:**

I assessed the sensibility of the derivations and theory.

**Review Assessment: Checking Correctness Of Experiments:**

I assessed the sensibility of the experiments.

**Review Assessment: Thoroughness In Paper Reading:**

I read the paper at least twice and used my best judgement in assessing the paper.

---

> ### Author Response · Authors · 2019-11-15
> **Response to Reviewer #1**
>
> We appreciate Reviewer 1 for the very helpful and constructive suggestions.  Following the suggestions, we have improved the paper to (1) emphasize the difference between the two types of loss corrections in Section 2.1, (2) include [1-2] into the related work section, and (3) add a training time analysis in Appendix D. Next we provide the computation time analysis.
>
> In terms of inference time, our method does not introduce extra computation if a single model (e.g. theta1) is used for test, whereas the inference time is doubled if we use the ensemble of both models.
> In terms of training time, Appendix D shows a detailed analysis. We first compare the total training time of DivideMix to several SOTA methods, using a single Nvidia V100 GPU. DivideMix (5.2 h) is slower than Co-teaching+ (4.3 h) but faster than P-correction (6.0 h) and Meta-Learning (8.6 h) which involve multiple training iterations. We also break down the computation time per-epoch for each operation in DivideMix, Co-Divide (Alg. 1, line 4-8) takes 17.2 seconds, MixMatch of data (Alg. 1, line 12-24) takes 16.0 seconds, whereas the model’s forward-backward computation (Alg. 1, line 25-27) takes 12.5 seconds.

---

### Official Review · AnonReviewer3 · 2019-10-25
**Official Blind Review #3**

**Rating:** 6

**Review:**

This paper proposes a DivideMix framework for learning with noisy labels, where they first Co-Divide the training data into a labeled clean set and an unlabeled noisy set by modeling the per-sample loss distribution with GMM and using the small loss trick, then they exploit MixMatch to train the model on those labeled and unlabeled data in a semi-supervised manner. Experiments and comparisons with SOTA are provided, together with an ablation study.

Pros:
-The paper bridges the area of learning with noisy labels with semi-supervised learning and proposes an interesting method to learn with noisy labels in a semi-supervised manner. The treatment proceeds from analyzing good unsupervised loss functions, improving the MixMatch and implementing thorough experiments.

-The paper is clear and flows smoothly. The treatment is thorough, proceeding from designing algorithms, implementing experiments and an ablation study.

-The impact of the method is a clear asset. Most existing label noise works focus on certain noise models, but this paper is more general and proposes an interesting method to learn with noisy labels in a semi-supervised manner, and the experimental results are promising.

-The effort made on designing the confidence penalty for asymmetric noise is interesting.

Remarks:
-Sec 3.1: it seems that \tau is an important hyperparameter for dividing the noisy data into labeled and unlabeled sets, but in Sec 4 I only see that \tao is set to be 0.5 or 0.6 for CIFAR, what about other experiments? I’m also wondering if \tao needs some decay during training? For example, it may be 0.5 at early training epochs, but may be smaller at last, since deep NNs are gradually fitting noisy features during training.

-Algorithm: the proposed method needs to train two networks simultaneously. During each epoch, it firstly divides the noisy data by modeling the per-sample loss distribution with GMM, and then do MixMatch with label co-refinement and co-guessing. I’m a bit concerned about efficiency. So how about the computation time?

-Experiments: more details on experimental protocol may be needed: what kind of hyperparameter tuning was done? How many repeated runs? It would be helpful to report the means and standard deviations based on repeated samplings.

Overall take: this paper proposes a thorough treatment of learning with noisy labels in a semi-supervised manner, designing the algorithm and testing it empirically, which is an interesting and important contribution. My only concern is about the novelty since the small loss trick in label noise and the MixMatch approach in SSL are already explored by many recent studies, but to the best of my knowledge, this paper is the first to unify them to solve label noise problems.

**Experience Assessment:**

I have published one or two papers in this area.

**Review Assessment: Checking Correctness Of Derivations And Theory:**

I assessed the sensibility of the derivations and theory.

**Review Assessment: Checking Correctness Of Experiments:**

I assessed the sensibility of the experiments.

**Review Assessment: Thoroughness In Paper Reading:**

I read the paper at least twice and used my best judgement in assessing the paper.

---

> ### Author Response · Authors · 2019-11-15
> **Response to Reviewer #3**
>
> We appreciate Reviewer 3 for the recognition of this paper and the valuable comments. Next we response to the questions raised by the reviewer.
>
> Question 1: What is the value of \tau in other experiments?
> Response: As explained in Appendix B, we set the value of \tau as 0.5 for other experiments.
>
> Question 2: I’m also wondering if \tau needs some decay during training, since deep NNs are gradually fitting noisy features during training.
> Response: We find that the proposed DivideMix can prevent the network from fitting to label noise and keep higher loss for noisy samples. This is supported by Figure 3 in Appendix, which shows that the GMM improves in distinguishing clean and noisy samples as training proceeds. Therefore, we can keep a constant \tau during training.
>
> Question 3: I’m a bit concerned about efficiency. So how about the computation time?
> Response: In terms of inference time, our method does not introduce extra computation if a single model (e.g. theta1) is used for test, whereas the inference time is doubled if we use the ensemble of both models.
> In terms of training time, we have added a training time analysis in Appendix D. We first compare the total training time of DivideMix to several SOTA methods, using a single Nvidia V100 GPU. DivideMix (5.2 h) is slower than Co-teaching+ (4.3 h) but faster than P-correction (6.0 h) and Meta-Learning (8.6 h) which involve multiple training iterations. We also break down the computation time per-epoch for each operation in DivideMix, Co-Divide (Alg. 1, line 4-8) takes 17.2 seconds, MixMatch of data (Alg. 1, line 12-24) takes 16.0 seconds, whereas the model’s forward-backward computation (Alg. 1, line 25-27) takes 12.5 seconds.
>
> Question 4: More details on experimental protocol may be needed: What kind of hyperparameter tuning was done? How many repeated runs?
> Response: As explained in Section 4.1 and Appendix B, we keep most hyperparameters fixed across experiments and do light tuning on \lambda_u (weight for unsupervised loss) on a per-experiment basis using a small validation set. We report results on single runs due to time concerns with the large number of experiments that we have conducted. Since all methods use the same training data, we believe that the results are fair for comparison. We thank the reviewer for this suggestion and would perform repeated runs in the future.

---

> > ### Comment · AnonReviewer3 · 2019-11-15
> > **Thank you**
> >
> > The authors have responded to my questions, and I have no other comments to make.

---

### Public Comment · ~Sara_Sabour1 · 2019-09-30
**IOU of labeled set during training**

Thank you for your well written and thorough study. The joint training was specially interesting to me. Have you looked at the intersection over union of the samples in one set (labeled) during training of the two networks?

This could be a divergence indicator of the two networks. If iou is high, maybe one can enforce (guarantees even) divergence with aux losses. But if analyzing the iou plot shows they are already sufficiently diverged there would be no point in it.

Thanks

---

> ### Author Response · Authors · 2019-10-02
> **Divergence analysis**
>
> Thanks for your comments!
> We have indeed analyzed the divergence of the two networks using the networks' prediction discrepancy (on the same images) as an indicator. Our result shows that the two networks can stay sufficiently diverged because of multiple factors: different parameter initialization, different training data division, different mini-batch sequence, and different training targets. The iou of the labeled set is one of the causes, rather than a direct indicator of the network divergence. The iou would get smaller as training proceeds because both networks would get better at dividing clean and noisy samples.

---

### Public Comment · ~Vikas_Verma1 · 2019-12-30
**Relevant paper**

Hi,

It is quite interesting paper!

Since your work is closely based on MixMatch,  I would like to point out to our previous work ICT, which is a precursor of MixMatch.

https://www.ijcai.org/proceedings/2019/0504.pdf

---

### Public Comment · ~Xinshao_Wang1 · 2020-08-24
**Algorithm 1 seems very complex.**

I have read this work carefully. However, it seems to me that *Algorithm 1 is extremely complex*:

1. Two networks are required to train;
2. A warm-up stage (standard training) is required to train each network. How many iterations should we train? How to determine this?
3. In every epoch, we need to divide the whole dataset into two subsets (labelled and unlabelled subsets). Therefore, every epoch will *take much more time than standard training.*
  3.1 Two Gaussian Mixture Models are trained;
  3.2 Two networks: num_iters \times 2;
  3.3 Mixup data augmentation;

Consequently, here, I would like to share some much simpler methods:
1.  IMAE for Noise-Robust Learning: Mean Absolute Error Does Not Treat Examples Equally and Gradient Magnitude's Variance Matters
https://arxiv.org/abs/1903.12141
2. Derivative Manipulation for General Example Weighting
https://arxiv.org/abs/1905.11233
3. Progressive Self Label Correction (ProSelfLC) for Training Robust Deep Neural Networks
https://xinshaoamoswang.github.io/blogs/2020-06-07-Progressive-self-label-correction/

https://xinshaoamoswang.github.io/blogs/2020-06-14-Robust-Deep-LearningviaDerivativeManipulationIMAE/

---

> ### Author Response · Authors · 2020-08-25
> **Thanks but DivideMix is a simple method.**
>
> Hi Xinshao,
>
> Thanks for your comments and sharing of your papers. We would like to note that the techniques used in DivideMix (i.e. warm-up, co-training, mixup) are commonly adopted in many previous works. Rather than being complex, DivideMix is a simple method that is easy to use.

---

> > ### Public Comment · ~Xinshao_Wang1 · 2020-08-25
> > **About warm-up, co-training**
> >
> > Dear authors,
> >
> > I hope all of you are very well. Many thanks for your kind reply.
> > I totally understand that differnt persons have different appetite. Presumably,  it is simple in the context of co-training and warm-up training.
> >
> > Very personally, I am not in favor of warm-up training, co-training due to those questions
> > (those questions are for warm-up training and co-training, not specifically for this work, i.e. DivideMix ):
> > 1. How many iterations should we train in warm-up stage? How to determine this?  Will  warm-up stage overfit?
> > 2. Why two networks since by SGD (sometimes Dropout), one network can be interpreted as an ensemble model.  Or why not three, or even many more networks?
> >
> > Specifically for this work,  in every epoch, DivideMix divides the whole dataset into two subsets (labelled and unlabelled subsets). Therefore, it seems to me that DivideMix is complex, and not scalable to very large datasets.
> >
> > Finally, please kindly let me know if I am wrong.
> >
> > Many thanks.

---

### Decision · Program_Chairs · 2019-12-19

**Decision:**

Accept (Poster)

**Comment:**

This paper proposes an algorithm for noisy labels by adopting an idea in the recent semi-supervised learning algorithm.

As two problems of training noisy labels and semi-supervised ones are closely related, it is not surprising to expect such results as pointed out by reviewers. However, reported thorough experimental results are strong and I think this paper can be useful for practitioners and following works.

Hence, I recommend acceptance.